# Genetic Study of SARS-CoV-2 Non Structural Protein 12 in COVID-19 Patients Non Responders to Remdesivir

Marta Santos Bravo,[a,b] Rodrigo Alonso,[c] Dafne Soria,[a] Sonsoles Sánchez Palomino,[d] Ángela Sanzo Machuca,[e] Cristina Rodríguez,[a] José Alcamí,[d,f,g] Francisco Díez-Fuertes,[f,g] Àlvar Simarro Redon,[a] Juan Carlos Hurtado,[a,b] Francesc Fernández Avilés,[h] Marta Bodro,[i] Elisa Rubio,[a] Jose Luis Villanueva,[a] Andrea Vergara,[a] Pedro Castro,[j] Montserrat Tuset,[k] Genoveva Cuesta,[a] Pedro Puerta,[c] Carolina García,[c] María del Mar Mosquera Gutiérrez,[a] Miguel J. Martínez,[a,b,g] Jordi Vila,[a] Alex Soriano,[c] María Ángeles Marcos[a,b]

aDepartment of Microbiology, Hospital Clinic of Barcelona, Barcelona, Spain
bInstitut of Global Health of Barcelona (ISGlobal), Barcelona, Spain
cDepartment of Infectious Diseases, Hospital Clinic of Barcelona, Barcelona, Spain
dAIDS Research Group, Institut de Recerca Biomèdica August Pi i Sunyer (IDIBAPS), Barcelona, Spain
eUniversitat Autónoma de Barcelona, Barcelona, Spain
fAIDS Immunopathogenesis Unit, Instituto de Salud Carlos III, Madrid, Spain
gCIBER de Enfermedades Infecciosas (CIBERINFEC), Instituto de Salud Carlos III, Madrid, Spain
hHematological Unit, Hospital Clinic of Barcelona, Barcelona, Spain
iTransplant Unit, Hospital Clinic of Barcelona, Barcelona, Spain
jIntensive Care Unit, Hospital Clinic of Barcelona, Barcelona, Spain
kPharmacology Unit, Hospital Clinic of Barcelona, Barcelona, Spain

Alex Soriano and María Ángeles Marcosa are co-principal investigators.

**ABSTRACT** Remdesivir (RDV) was the first antiviral drug approved by the FDA to treat severe coronavirus disease-2019 (COVID-19) patients. RDV inhibits SARS-CoV-2 replication by stalling the non structural protein 12 (nsp12) subunit of the RNA-dependent RNA polymerase (RdRp). No evidence of global widespread RDV-resistance mutations has been reported, however, defining genetic pathways to RDV resistance and determining emergent mutations prior and subsequent antiviral therapy in clinical settings is necessary. This study identified 57/149 (38.3%) patients who did not respond to one course (5-days) ($n = 36/111$, 32.4%) or prolonged (5 to 20 days) ($n = 21/38$, 55.3%) RDV therapy by subgenomic RNA detection. Genetic variants in the *nsp12* gene were detected in 29/49 (59.2%) non responder patients by Illumina sequencing, including the *de novo* E83D mutation that emerged in an immunosuppressed patient after receiving 10 + 8 days of RDV, and the L838I detected at baseline and/or after prolonged RDV treatment in 9/49 (18.4%) non responder subjects. Although 3D protein modeling predicted no interference with RDV, the amino acid substitutions detected in the nsp12 involved changes on the electrostatic outer surface and in secondary structures that may alter antiviral response. It is important for health surveillance to study potential mutations associated with drug resistance as well as the benefit of RDV retreatment, especially in immunosuppressed patients and in those with persistent replication.

**IMPORTANCE** This study provides clinical and microbiologic data of an extended population of hospitalized patients for COVID-19 pneumonia who experienced treatment failure, detected by the presence of subgenomic RNA (sgRNA). The genetic variants found in the *nsp12* pharmacological target of RDV bring into focus the importance of monitoring emergent mutations, one of the objectives of the World Health Organization (WHO) for health surveillance. These mutations become even more crucial as RDV keeps being prescribed and new molecules are being repurposed for the treatment of COVID-19. The present article offers new perspectives for the clinical management of non responder

Address correspondence to Marta Santos Bravo, martasantosbravo@gmail.com.

The authors declare no conflict of interest.

patients treated and retreated with RDV and emphasizes the need of further research of the benefit of combinatorial therapies and RDV retreatment, especially in immunosuppressed patients with persistent replication after therapy.

**KEYWORDS** COVID-19, remdesivir, resistance mutations, subgenomic RNA, retreatment, SARS-CoV-2, genetic variants

The global pandemic of novel coronavirus disease-2019 (COVID-19) caused by severe acute respiratory syndrome coronavirus-2 (SARS-CoV-2) has created an urgent effort to repurpose antiviral inhibitors to control viral replication and improve clinical outcomes (1). Remdesivir (RDV) was originally developed in response to the 2014–2016 Ebola outbreak in West Africa (2) and has shown broad-spectrum activity *in vitro* and *in vivo* against pathogenic human coronaviruses, including the novel SARS-CoV-2 (3). RDV was the first drug to be approved by the FDA in October 2020 and has been extensively used in clinical practice during the COVID-19 pandemic in hospitalized patients (4). Recently, two oral prodrugs have also been approved for treatment of COVID-19 patients: molnupiravir and nirmatrelvir/ritonavir (5, 6).

RDV, formerly GS-5734, is a nucleoside analog prodrug that inhibits the non structural protein 12 (nsp12) subunit of the RNA-dependent RNA polymerase (RdRp) by competing with its usual natural substrate ATP (7). The nucleoside analog is incorporated into the generating RNA strand and evades proofreading to successfully inhibit viral RNA synthesis. Several clinical trials and a recent case-control study demonstrated reduced time to recovery, hospitalization time, morbidity, and mortality (8–10).

One of the major concerns for health surveillance is determining emergent mutations that could be associated with drug resistance, fitness advantage, immune escape, or better adaption to the host. Only a few studies have attempted to characterize amino acid substitutions that could confer resistance to RDV by *in silico* prediction, *in vitro*, or in animal models (7, 11–15). For instance, F480L, V557L, and E802D have been described as conferring 2.4-, 5.2-, and 6-fold decreased sensitivity to RDV *in vitro*, respectively (7, 14). Moreover, E802D was recently detected in an immunocompromised patient after RDV therapy (15); however, no other resistant clinical cases or evidence of global widespread transmission of RDV-resistant mutants have been described, after treating with RDV for over a year.

This study aimed to identify novel genetic variations in the *nsp12* gene in clinical samples before and after RDV therapy in severe COVID-19 patients who did not respond to therapy with RDV.

## RESULTS

In a cohort of 149 patients hospitalized for COVID-19 pneumonia, 111 received a 5-dose course of RDV (5d-RDV), and 38 received prolonged RDV therapy (>5d-RDV) (Fig. 1). The median (quartile 1 [Q1], Q3) age of the 5d-RDV subset was 62.7 (54, 73) years old, 46 (41.4%) were female, 81 (73%) presented with at least one comorbidity, 34 (30.6%) were admitted to the intensive care unit (ICU), and 7 (6.3%) died from all-cause mortality (Table 1). The median (Q1, Q3) age in the >5d-RDV subgroup was 59 (53, 67) years old, and they were treated with RDV 10 (7, 21) days. Of the 38 patients, 12 (32.6%) were female, 33 (86.8%) had at least one comorbidity, 19 (50%) were admitted to the ICU, and 8 (21.1%) died from all-cause mortality, but 3 deaths were not related to COVID-19. Extended clinical data of both subsets are shown in Table 1.

SARS-CoV-2 subgenomic RNA (sgRNA) was used to identify 57/149 (38.3%) patients with actively replicating virus after treatment, classified as non responders (Fig. 1). There were 36/57 (63.2%) non responders in the 5d-RDV subgroup, and 21/57 (36.8%) from the >5d-RDV subgroup. The 17 remaining patients in the >5d-RDV cohort were not tested for sgRNA and were excluded because the pre- or post treatment swab could not be recovered. Cycle threshold ($C_T$) values corresponding to the genomic RNA (gRNA) real-time reverse transcriptase PCR (RT-PCR) were maintained or decreased

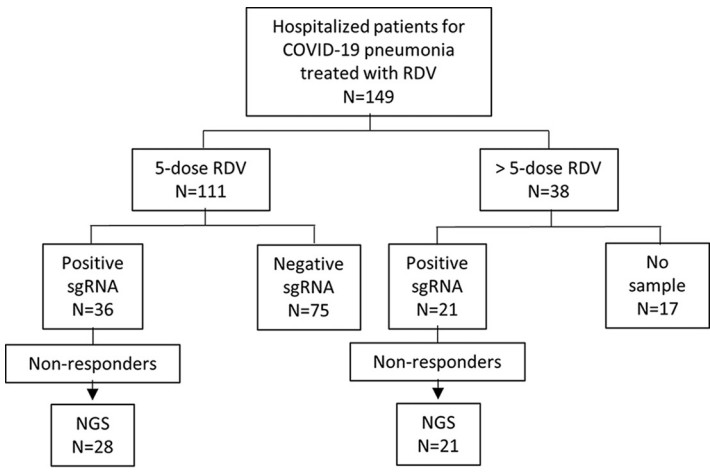

**FIG 1** Diagram of the study design. Patients positive for SARS-CoV-2 who were admitted to the Hospital Clinic of Barcelona (Spain) for COVID-19 pneumonia and were treated with remdesivir (RDV) were included in the study. They were treated with 5 doses (1 course) of RDV or with longer treatments (>5 doses). Subgenomic RNA (sgRNA) detection was performed on all samples in order to detect viral replication before and after treatment. Positive and negative sgRNA in the diagram indicate the results of the sample after the last day of RDV treatment. Patients with positive sgRNA after treatment were classified as non responders and were sequenced by next-generation sequencing (NGS). Not all clinical samples could be sequenced with high enough quality to be analyzed and included in the study due to RNA degradation or low viral load in the sample. Of the samples from the >5-dose subpopulation that could be studied, all were sgRNA RT-PCR positive in the last sample.

during treatment in non responder patients. Nucleotide changes in the *nsp12* gene were detected by next-generation sequencing (NGS) and were compared before and after RDV therapy in non responder patients. Sequencing quality was achieved in both samples in 28/34 patients of the 5d-RDV subset and in all 21 patients of the >5d-RDV subset.

NGS allowed the detection of 18 *nsp12* nucleotide substitutions in 17/28 (60.7%) patients from the 5d-RDV subgroup and in 12/21 (57.1%) patients receiving prolonged RDV therapy. Of the 18 nucleotide substitutions identified, 9 conferred an amino acid

**TABLE 1** Clinical characteristics of the study population according to the treatment of remdesivir received[a]

| Clinical characteristic | 5d-RDV | >5d-RDV |
|---|---|---|
| No. of participants | 111 | 38 |
| Median (IQR) age (yrs) | 62.7 (54–73) | 59 (56–67) |
| No. (%) of female participants | 46 (41.4) | 12 (32.4) |
| Days of RDV therapy (median [IQR]) | 5 | 10 (7–21) |
| | | |
| No. (%) of participants with comorbidities | | |
| Total | 81 (73) | 33 (86.8) |
| Hypertension | 51 (45.9) | 8 (21.1) |
| Diabetes mellitus | 26 (23.4) | 5 (13.2) |
| Obesity | 18 (16.2) | 3 (7.9) |
| Cardiovascular disease | 32 (28.8) | 3 (7.9) |
| Chronic pulmonary disease[b] | 24 (21.6) | 6 (15.8) |
| Chronic kidney failure | 9 (8.1) | 1 (2.6) |
| Hematological malignancy[c] | 15 (13.5) | 25 (65.8) |
| Solid malignancy with active chemotherapy | 4 (3.6) | 1 (2.6) |
| Transplant recipient | 4 (3.6) | 6 (15.8) |
| Other disorder treated with immunosuppressors | 7 (6.3) | 1 (2.6) |
| | | |
| No. (%) of participants with: | | |
| ICU admission | 34 (30.6) | 21 (55.3) |
| Mortality | 7 (6.3) | 8 (21.1) |

[a]RDV, remdesivir; IQR, interquartile range; ICU, intensive care unit.
[b]Chronic pulmonary disease includes chronic obstructive pulmonary disease and asthma.
[c]Hematological malignancy includes lymphoma or leukemia.

**TABLE 2** Non structural protein (nsp12) nucleotide substitutions detected in clinical isolates[a]

| Mutation[b] | Location | No. of participants with mutation (days of treatment)[c] | | | No. of participants with: | |
|---|---|---|---|---|---|---|
| | | Pre-RDV | Pre-/post-RDV | Post-RDV | ICU admission | Mortality[d] |
| A13535G (Y32C) | | 2 | 1 (10) | 1 (5) | 1 | |
| C13551T | | | | 1 (18) | 1 | 1 |
| G13564T (V42L) | | | 1 (5) | | | |
| A13689T (E83D) | | | | 1 (18) | 1 | 1 |
| A13711G (K91G) | | | 1 (5) | | | |
| C14119T | Nsp8 interaction | | 1 (5) | | | |
| C14120T (P227L) | Nsp8 interaction | | 3 (5 [n = 2], 10 [n = 1]) | 1 (5) | 3 | 1 |
| C14178T | Nsp8 interaction | | 1 (5) | | | |
| C14547A | Nsp7-8 interaction | | 2 (5) | 3 (5) | 3 | |
| C14703T (T422I) | RNA binding site (motif G) | | 1 (10) | | 1 | |
| C15237T | RNA binding site (motif B) | | | 1 (10) | | |
| C15240T | RNA binding site (motif B) | | 1 (5) | | 1 | 1 |
| C15324T | RNA binding site | 2 | 1 (30) | 2 (5) | 3 | 1 |
| C15441T | RNA binding site (motif C) | | 1 (5) | | | |
| G15627T (E729D) | RNA binding site (motif E) | | 1 (5) | | 1 | |
| G15652T (D738Y) | RNA binding site | | 1 (5) | 1 (5) | 1 | |
| G15910T | | | 1 (10) | | | |
| C15952A (L838I) | | | 8 (10 [n = 7]; 21 [n = 1]) | 1 (20) | 6 | 3 |

[a]Nsp, non structural protein; RDV, remdesivir; ICU, intensive care unit.
[b]The nucleotide position is indicated with respect to the Wuhan-Hu-1 reference genome (GenBank accession number MN908947.3). The amino acid substitution is indicated in parentheses.
[c]Number of subjects with the specific substitution at baseline (pre-RDV), at baseline and after treatment with remdesivir (pre-/post-RDV), and after (post-RDV) therapy. The number of days of treatment received when the mutation was found is indicated in parentheses.
[d]Number of participants who died due to all-cause mortality.

substitution, pointing to genetic evolution after treatment (Table 2). Mutations were either detected at baseline (pre-RDV), before and after treatment (pre-/post-RDV), or emerged *de novo* after treatment (post-RDV). The only non synonymous mutation detected after treatment was E83D. E83D emerged in a patient (patient 7; see Table S1 in the supplemental material) with diffuse large B-cell lymphoma that was treated with R-CHOP combination chemotherapy, with several antiviral drugs (lopinavir, ritonavir, hydroxychloroquine, azithromycin, RDV) and convalescent plasma. The E83D mutation emerged *de novo* after 2 courses (10 and 8 days) of RDV. Ultimately, this patient was admitted to the ICU and died after 8 months of SARS-CoV-2 infection. The most frequent mutation (18.4%) was L838I, found at baseline and after therapy in 8 patients, and in 1 subject only after prolonged treatment. Clinical and genetic data of all patients with *de novo* mutations is shown in Table S1.

NGS allowed the determination of SARS-CoV-2 lineages. The Alpha (B.1.1.7) variant was mainly detected in subjects who underwent 5d-RDV ($n = 19$) rather than >5d-RDV ($n = 3$), whereas Delta (B.1.167) was more prominent in >5d-RDV subjects ($n = 12$) than in the 5d-RDV subgroup ($n = 4$). Variant B.1.177 emerged as an outbreak in Spain during the summer of 2020, and its incidence was similar in both groups (5d-RDV, $n = 5$; >5d-RDV, $n = 6$).

**Predicting the phenotype of novel *nsp12* mutations by 3D protein modeling.** The position of all genetic variants detected at any point during the treatment were located in the gene structure of the *nsp12* (Fig. 2) and non synonymous mutations in a 3D protein structure of the RdRp (Fig. 3A). *In silico* studies have postulated that RDV binds strongly to the active pocket of the nsp12 by electrostatic interaction with residues R553, R555, T556, K551, W617, D618, Y619, D623, S682, N691, D760, D761, A762, W800, E811, F812, K813, and S814 and by van der Waals bonds in K621, K622, D623, and L758 (11, 12). Our genotypic results showed the emergence of mutations located next to residues involved in RDV-nsp12 binding (E729D, D738Y, L838I), in the nsp8-nsp12 interaction region (P227L), and inside the RNA-binding domain of the polymerase (T422I). However, RdRp 3D protein modeling shows that all novel amino acid substitutions are distant from the RNA binding domain, where RDV inhibits polymerase activity. Replacing one amino acid of a helix for another can change the repartition of amino

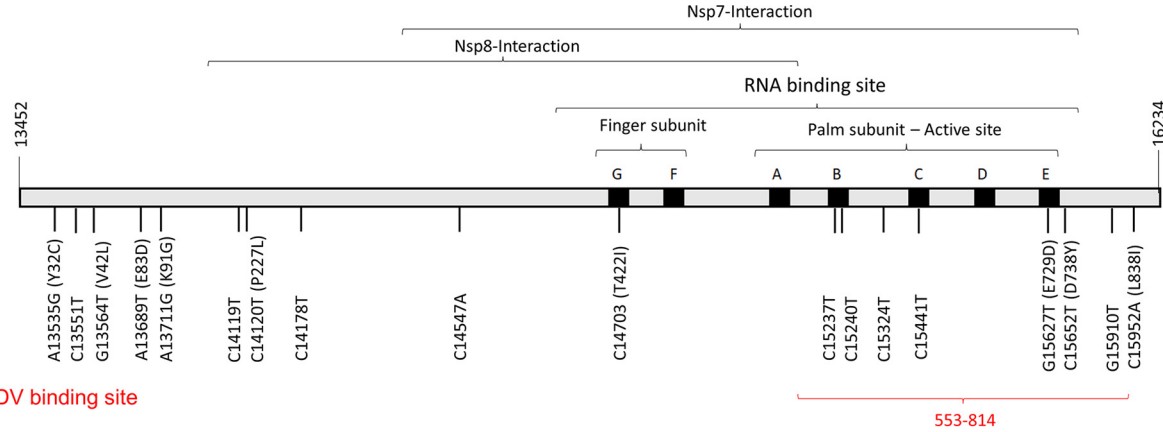

**FIG 2** Gene structure of SARS-CoV-2 non structural protein 12 with novel mutations detected. The nucleotide position is indicated with respect to the Wuhan-Hu-1 reference genome (GenBank accession number MN908947.3). Amino acid substitutions are indicated in parentheses. The gene structure is based on the work of Gao et al. (25).

acids exposed to solvent, as found for E729D (Fig. 3B). Slight changes in conformation were detected for the remaining amino acid substitution.

Most mutations changed negatively charged amino acids to aliphatic-chained ones, substituting the highly negative electrostatic outer surface of nsp12 for the major over-all neutral status of the surface (Fig. 3C). Changes in polarity, charge, and size of the amino acids could potentially modify interactions between nsp12 and antivirals.

## DISCUSSION

The study of sgRNA allowed the identification of 57/149 (38.3%) patients who did not respond to 5 days or >5 days of RDV therapy. Eighteen genetic variants in the *nsp12* gene were detected by NGS in 29/49 (59.18%) non synonymous subjects. No significant viral mutations were determined to be associated with the failure of RDV treatment, except for the *de novo* E83D mutation, which emerged after 18 days of RDV. In addition, the L838I mutation was found at found after prolonged treatment in 1 patient and before and after RDV in 8 non responders (18.4%). Its localization adjacent to the previous E802D RDV-resistant mutation makes it attractive for further studies.

Genetic variants in *nsp12* are frequently favored by prolonged replication and antiviral exposure; however, *nsp12* mutations arose at similar rates after 5 days or longer of RDV treatment, independently of the duration of therapy (5d-RDV, 60.7%; >5d-RDV, 57.1%). Surprisingly, 5d-RDV subjects presented more *de novo* mutations ($n = 6$) than did >5d-RDV subjects ($n = 3$). Almost all mutations were detected at baseline in at least one subject, showing an evolutionary tuning of the viral proteins to a new host, although a response for antiviral selective pressure cannot be excluded in those for whom disease persisted after treatment. In the nonresponder patients, the viral loads progressively increased during treatment, as the $C_T$ values were maintained/decreased and sgRNA remained detectable, which could be due to either slow viral shedding or treatment failure.

Despite the high frequency of substitutions in *nsp12*, none of them have been previously described to confer resistance to RDV (15). We detected the L838I mutant, nearby the E802D RDV-resistant mutation, in 8 patients before and after treatment and in 1 patient only post therapy. All of them received either 10, 20, or 21 days of RDV therapy, 6 of them were admitted to the ICU, and 3 died. Therefore, L838I seems to be associated with a worse prognosis, although there are not enough cases to confirm this conclusion. This suggests that this naturally occurring variant may provide improved viral escape for the inhibitor, since the template stalling action of RDV limits the emergence of spontaneous mutations (7, 16).

E83D emerged in the SARS-CoV-2 Delta variant, which infected an immunosuppressed patient who was admitted to the ICU and died after 9 months of infection. The

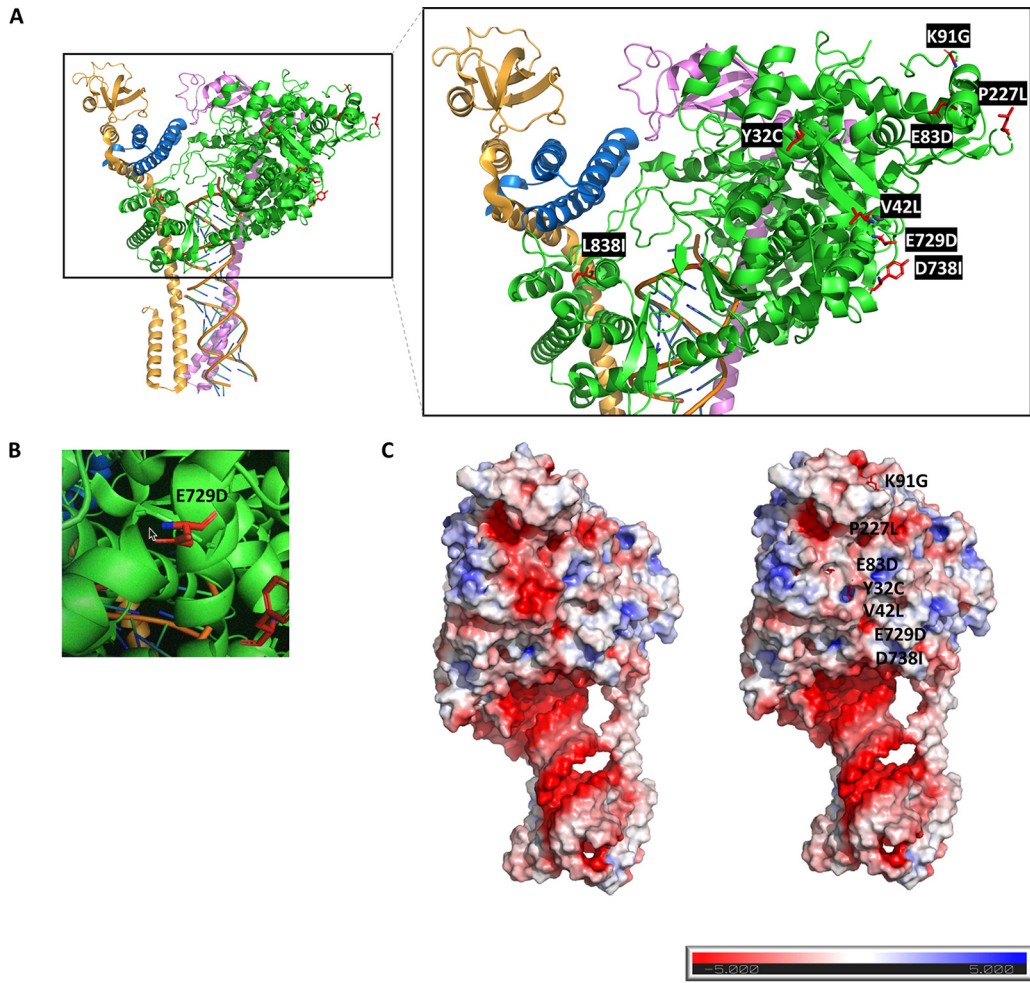

**FIG 3** Location of novel RdRp mutations in 3D protein models. (A) Theoretical structure of RdRp is represented in the ribbon structure of the cryo-EM model using PyMOL. The different subunits of the RdRp are indicated as follows: nsp7 in blue, nsp8a in yellow, nsp8b in pink, nsp12 in green. The RNA duplex is colored orange and blue. Novel genotypically detected mutations are indicated in red (right). (B) E729D mutation breaks the alpha-helix secondary structure of the nsp12. (C) APBS-generated electrostatic surface of the RdRp. Negatively charged areas are indicated in red and positively charged areas in blue. Mutations are visualized in red and labeled in the figure.

E83D mutation was only detected after 10 and 8 days of RDV treatment but not at baseline, making it a candidate for phenotyping. Using 3D protein modeling, no interference was predicted in the interaction with RDV; however, its implications for fitness advantage, inhibitor escape, and adaption to the host are unknown.

E729D and D738Y may also have implications for antiviral response, as they were located in the palm subunit of the polymerase active site next to the residues involved in the RDV-nsp12 interaction, and E729D also modified an alpha-helix structure. T422I is located in the conserved motif G, which controls RNA template attachment. It is close to the resistant mutations previously described *in vitro* (F480L, V557L) and *in vivo* in an immunocompromised patient with persistent viremia (D484Y) (7, 17). They do not alter the RdRp catalytic site but are thought to impact the RdRp fidelity checking step before catalysis. Molecular surveillance of this region in RDV-treated COVID-19 patients is suggested to be warranted (16). Even though 3D protein modeling predicted that none of the mutations found in this study block the binding pocket of RDV, they involved changes to the electrostatic outer surface and to secondary structures that may alter the antiviral response.

Lineage could be a concern for the severity of the disease and the antiviral response. The Alpha variant was more frequent in the 5d-RDV group and Delta in the

>5d-RDV group, which agrees with its higher pathogenicity. Besides the possible bias caused by the time of inclusion of the patients, the Delta variant could have caused a worse response to RDV and influenced the need for a second course of treatment.

The available literature about RDV retreatment has reported only a few clinical cases (18, 19), but no large studies have been carried out, with the result that this practice is still unaddressed in the current treatment guidelines (20). However, this study provides further information on the response to RDV treatment and retreatment in patients whose active replication was previously checked by sgRNA and for whom clinical data were collected.

In conclusion, no significant virological resistance was determined after different courses of RDV in non responder severe COVID-19 patients, and the duration of RDV treatment does not seem to be a risk factor for developing RDV resistance mutations. However, the mutations found in this study, especially E83D, have potential for further evaluation by recombinant phenotyping. It is crucial to monitor antiviral resistance, as one of the health surveillance objectives of the World Health Organization (WHO), and to study the potential benefit of combinatorial therapies and RDV retreatment, especially in immunosuppressed patients or those with persistent replication.

## MATERIALS AND METHODS

**Study population.** This was an observational prospective study that included 149 COVID-19 patients admitted to the Hospital Clinic of Barcelona (Spain) between February 2021 and November 2021, who met the criteria for receiving RDV according to the recommendations of the Spanish Medicine Agency. These criteria were the following: (i) SARS-CoV-2 positivity confirmed by RT-PCR, (ii) ≤7 days from onset of symptoms, (iii) radiological signs of pneumonia, and (iv) requiring supplemental oxygen support, or a respiratory rate of ≥24 breaths per minute, or an arterial oxygen partial pressure ($PaO_2$)/fractional inspired oxygen ($FiO_2$) ratio of <300 mm Hg. An RDV dose of 200 mg was used as a loading dose the first day, and 100 mg/24 h was given for the next 4 consecutive days. Some immunosuppressed patients received prolonged remdesivir therapy (>5d-RDV), a decision made by the physician in charge according to the clinical evolution and the immune status of the patient.

Nasopharyngeal/throat swabs were collected before the first RDV dose and after the last dose for each patient. Both samples were tested for SARS-CoV-2 genomic RNA (gRNA) and subgenomic RNA (sgRNA) by real-time reverse transcriptase PCR (RT-PCR). Although there is some controversy concerning the use of sgRNA to detect active viral replication, we previously validated this technique with a viral culture (21). Nonresponder subjects were considered those in whom sgRNA was detected at the end of the RDV treatment. Nucleotide changes were determined by next-generation sequencing (NGS) using the Illumina platform in pre- and post-RDV treatment clinical samples of the nonresponders. Novel amino acid substitutions found in the clinical isolates were evaluated using a 3D modeling structure of the protein *in silico*. This study design is shown in Fig. 1.

Clinical data were collected to study the overall population, considering the following variables: age, significant comorbidities, days of treatment with RDV, intensive care unit (ICU) admission, and all-cause mortality.

**RT-PCR for genomic and subgenomic RNA detection for SARS-CoV-2.** The presence of SARS-CoV-2 gRNA was determined by real-time RT-PCR using the automated Cobas 6800 system (Roche, Barcelona) according to the manufacturer's instructions.

Inactivation was performed using a 1:1 volume of Cobas Omni lysis reagent (Roche, Germany), and total nucleic acid extraction was conducted using the MagNA Pure compact system (Roche, Switzerland). Elutions were used for the sgRNA test and NGS. Envelope (E) sgRNA was detected by real-time RT-PCR following a previously described procedure (21). Throat/nasopharyngeal swabs and elutions were aliquoted and stored at −80°C until testing. Cycle threshold (Ct) values of >40 for gRNA or sgRNA RT-PCRs were considered negative.

**Next-generation sequencing of the complete SARS-CoV-2 genome.** The eluted SARS-CoV-2 RNA was reverse-transcribed into cDNA, and complete SARS-CoV-2 genome amplification was conducted following the openly available protocol developed by the ARTIC network (22) using the Illumina platform. The sequences obtained went through a bioinformatic pipeline based on the previously described open-source pipeline SARS-CoV2-Mapping, available on GitLab from FISABIO-NGS (23).

Mutations were considered in the generated consensus sequence if they were present in ≥80% of reads, reached a minimum quality of 20, and had a minimum depth per position of 30×. Mutations were identified by aligning the consensus sequence with the Wuhan-Hu-1 reference genome (GenBank accession number MN908947.3) using Nextclade v.1.14.0. The quality of the sequences was determined using the quality control metrics of this tool.

**Lineage identity.** The SARS-CoV-2 lineage was identified using Nextclade v.1.14.0 according to the amino acid replacement determinant of each variant, as classified by the SARS-CoV-2 Interagency Group (SIG) and the Centers of Disease Control and Infection (CDC) (24).

**Molecular modeling of mutations in the SARS-CoV-2 RNA-dependent RNA polymerase.** Mutations found genotypically in the nsp12 subunit of the RdRp were generated in the ribbon structure

of the cryo-electron microscopy (cryo-EM) model of the replicating SARS-CoV-2 polymerase complex (PDB accession number 6YY7) using PyMOL Molecular Graphics System v.2.5.2 (Schrödinger). This software was also used for structural visualization. The electrostatic plugin Adaptive Poisson-Boltzmann Solver (APBS) v.2.1 included in PyMOL was used for macromolecular electrostatic calculations and to display the results as a molecular electrostatic potential surface.

**Ethical approval.** The ethics committee of our institution accepted the protocol (HCB/2021/0080), and the patients included provided signed informed consent to participate in the study.

**Data availability.** Raw data is available at NCBI accession number BioProject PRJNA895554.

## SUPPLEMENTAL MATERIAL

Supplemental material is available online only.
**SUPPLEMENTAL FILE 1**, PDF file, 0.4 MB.

## ACKNOWLEDGMENTS

We thank Donna Pringle for English language editing.

This work was financed by a Gilead Sciences grant (IN-ES-540-6089) and CIBER Enfermedades Infecciosas (CIBERINFEC), Instituto de Salud Carlos III, Madrid, España (CB21/13/00081). This work was financed by *ad hoc* patronage funds for research on COVID-19 from donations from citizens and organizations to the Hospital Clínic de Barcelona—Fundació Clínic per a la Recerca Biomèdica.

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
