## [Reviewer comments · Microbiology Spectrum]

Microbiology Spectrum

Genetic study of SARS-CoV-2 nsp12 in non-responder COVID-19 patients to remdesivir

Marta Santos Bravo, RODRIGO ALONSO, Dafne Soria, Sonsoles Sánchez-Palomino, Ángela Sanzo-Machuca, Cristina Rodríguez, José Alcamí, Francisco Diez Fuertes, Àlvar Simarro Redon, Juan Hurtado, Francesc Fernández Avilés, Marta Bodro, Elisa Rubio Garcia, Jose Villanueva, Andrea VERGARA, Pedro Castro, Montse Tuset, Genoveva Cuesta, Pedro Puerta, Carolina García-Vidal, Mar Mosquera, Miguel Martínez, Jordi Vila, Alex Soriano, and M^a Angeles Marcos

Corresponding Author(s): Marta Santos Bravo, Microbiology Department, Hospital Clínic I Provincial de Barcelona, University of Barcelona. Institute for Global Health (ISGlobal), Barcelona, Spain.

Review Timeline:

Submission Date:	June 28, 2022
Editorial Decision:	July 24, 2022
Revision Received:	August 22, 2022
Accepted:	September 2, 2022

Editor: Abimbola Kolawole

Reviewer(s): Disclosure of reviewer identity is with reference to reviewer comments included in decision letter(s). The following individuals involved in review of your submission have agreed to reveal their identity: ASAAD MOHAMMED ATAA (Reviewer #1); Jonathan Daniel Hulse (Reviewer #3)

Transaction Report:

DOI: <https://doi.org/10.1128/spectrum.02448-22>

July 24, 2022

Dr. Marta Santos Bravo

Microbiology Department, Hospital Clínic I Provincial de Barcelona, University of Barcelona. Institute for Global Health (ISGlobal), Barcelona, Spain.

Microbiology, Hospital Clinic of Barcelona

Villarroel Street, 170.

Barcelona, Barcelona 08036

Spain

Re: Spectrum02448-22 (Genetic study of SARS-CoV-2 nsp12 in non-responder COVID-19 patients to remdesivir)

Dear Dr. Marta Santos Bravo:

Link Not Available

Sincerely,

Abimbola Kolawole

Journals Department
Reviewer comments:

Reviewer #1 (Comments for the Author):

Dear authors, I would like to thank you for your work. Your work is impressive in this section and gives us some important points about Remdesivir as an antiviral SARS-CoV-2. It might give scientists a new way to cure COVID-19.

These my comments:-

Lines (27-28) We need to define genetic and biochemical pathways to RDV resistance and emphasize the need for additional

studies to define the potential for emergence of these or other RDV resistance mutations in clinical settings. So you need to edit this paragraph.

In lines (33 and 36) You need to review numbers

In line (118) Why did not add a gender variable?

In lines (153-160) There is concern about the death rate, which is high compared to the number of patients who received RDV. Are there drug interactions or the effect of this drug on the chronic diseases of patients and old age? This should be clarified.

In lines (215-216) Can you explain this paragraph?

In lines (232-235) You need to add more information to clear up this confusion.

In line (401) Figure 2, you need to change the color of words to make them clear to the audience.

No statistical software was used to analyze the results.

Reviewer #2 (Comments for the Author):

Bravo and co-workers reported a clinical series of some 100+ covid-19 cases with remdesivir treatment and associated viral whole-genome NGS data. They reported that multiple mutations on the nsp12 (RdRp) gene were found but they were not on or in the vicinity of the known RdRp active site and concluded that no virological resistance was found during short (5-day) and longer (5+) courses of remdesivir therapy. Integrated clinical and virological data are important to the field. But this reviewer identified technical flaws in the study design that should be properly addressed before supporting the conclusion of the present work.

Major

1. This study was not a case-control study and there was no functional validation of the mutations in vitro or in cell line to disprove their association with remdesivir resistance. In addition, the authors have no attempts to provide evidence that those sequences collected with remdesivir therapy has mutation rate comparable to background level (i.e., without remdesivir treatment) to support their claims that those mutations are not remdesivir-driven. As the authors performed WGS, such data should be readily available for comparison with those already reported in the literature. This is critically needed before jumping to any conclusions currently based on homologous modelling only without any functional validation.
2. In retrospect, the authors should have collected samples on an ideally daily basis so that those from non-responders could also be sequenced and analysed as long as they remained viral RNA positive and could have served as a comparison group for mutation rate and de novo mutation appearance. The inclusion of another control group comprising COVID-19 patients without remdesivir treatment is also essential. For example, a similar random mutation patterns between treatment and no-treatment groups can indirectly indicate that the mutations were not associated with antiviral resistance. The key message is that proving and disproving association of mutations to antiviral resistance require the same par of evidence that the current study unfortunately lack.

Minor

3. Line 133: "retrotranscribed" should have read as "reverse-transcribed".
4. Lines 133-136: More technical descriptions on the NGS workflow would be very helpful. Simply saying "as previously described" is far from reader-friendly. At least the authors should provide name of the pipeline and key algorithm used in the analysis NGS data.
5. Bioinformatics pipeline and cut-off/ threshold used in the identification of purported mutations associated (or not associated) with remdesivir therapy needs to be clearly described. For example, sequences existed as quasispecies and did the mutations need to be present in 100% of the illumine reads in order to qualify as a mutation? If no, what was the selection criteria and the rationale behind?
6. Figure 1: the "Positive sgRNA" and "Negative sgRNA" labels should indicate that they referred to samples collected at the last day of remdesivir dosing.
7. Lines 157-160: What do the numbers in parenthesis refer to, IQR or range or something else? Please clarify.
8. The work would benefit from English-editing.

Reviewer #3 (Comments for the Author):

This is a very interesting paper and I feel that it adds to the body of work surrounding COVID-19. Here are some suggestions:

Line 27: Delete 'of', it should read "No evidence of global widespread 28 RDV-resistance mutations has been reported"

Line 28 - 30: Change wording from 'or' into 'to'. "Determining emergent mutations prior to..."

Line 32: Why is there a (63.2%) in the sentence. Needs clarification.

Line 33 -34: Clarify Next Generation Sequencing. 454 Pyrosequencing or Illumina, or some other type?

Line 40 - 42: Try not to use 'and' two times in one sentence. Substitute one of the 'and' for 'as well as'

Line 47: Capitalize Remdesivir or use RDV since it was defined earlier.

Line 49: Capitalize Remdesivir or use RDV since it was defined earlier.

Line 79: Capitalize the drug names. They are proper nouns.

Line 89: Add a comma after in vitro

Line 90: Add a comma after V557L

Line 116: Clarify Next Generation Sequencing. 454 Pyrosequencing, Illumina, or some other type?

Line 157: Clarify what (54;73) means. Is this 54 - 73 years old?

Line 178: Remove the second 'only' in the sentence. It should read, "The only non-synonym mutation detected..."

Line 181: Try to avoid using 'and' two times in one sentence. Change the 'and' to 'as well as'

Line 194: Be constant with your italics of nsp12.

Line 196: Be constant with your italics of nsp12.

Line 195-198: Run-on sentence. Break into two sentences if possible.

Line 207: Be constant with your italics of nsp12.

Line 209: Be constant with your italics of nsp12.

Line 213 - 217: Avoid using 'and' multiple times in a sentence. Break this sentence into 1-3 sentences because it is too long.

Line 410: Make sure that in Table 1, all of your Clinical Characters are Capitalized. Be consistent.

Staff Comments:

Preparing Revision Guidelines

Please return the manuscript within 60 days; if you cannot complete the modification within this time period, please contact me. If you do not wish to modify the manuscript and prefer to submit it to another journal, please notify me of your decision immediately so that the manuscript may be formally withdrawn from consideration by Microbiology Spectrum.

**Genetic study of SARS-CoV-2 nsp12 in non-responder COVID-19 patients to remdesivir**

Marta Santos Bravo ^{1,2}, Rodrigo Alonso ³, Dafne Soria ¹, Sonsoles Sánchez Palomino ⁴, Angela
Sanzo Machuca ⁵, José Alcamí ^{4,6,7}, Francisco Díez ^{6,7}, Francesc Fernandez Aviles ⁸, Marta Bodro ⁹,
Elisa Rubio ¹, Jose Luis Villanueva ¹, Andrea Vergara ¹, Pedro Puerta ³, Carolina García ³, María del
Mar Mosquera Gutierrez ¹, Miguel J Martínez ^{1,2,7}, Alex Soriano ^{*3}, María Ángeles Marcos ^{*1,2}.

- 1. Department of Microbiology, Hospital Clinic of Barcelona, Spain.
2. Institut of Global Health of Barcelona (ISGlobal), Barcelona, Spain
3. Department of infectious diseases. Hospital Clinic of Barcelona, Spain
4. AIDS Research Group, Institut de Recerca Biomèdica August Pi i Sunyer (IDIBAPS),
Barcelona, Spain.
5. Universitat Autònoma de Barcelona, Barcelona, Spain.
6. AIDS Immunopathogenesis Unit. Instituto de Salud Carlos III, Madrid, Spain
7. CIBER de enfermedades Infecciosas (CIBERINFEC). Instituto de Salud Carlos III. Madrid,
Spain
8. Hematological Unit, Hospital Clinic of Barcelona, Spain.
9. Transplant Unit, Hospital Clinic of Barcelona, Spain.

**Corresponding author:**

Marta Santos Bravo, PhD.

martasantosbravo@gmail.com

Department of Clinic Microbiology, Hospital Clínic of Barcelona – University of Barcelona

ISGlobal Barcelona Institute for Global Health (Barcelona, Spain).

Villarroel Street, 170. Stairs 11, Floor 5th. 08036 Barcelona, Spain.

**ABSTRACT**

Remdesivir (RDV) was the first antiviral drug approved by the FDA to treat severe COVID-19
patients. RDV inhibits SARS-CoV-2 replication by stalling the non-structural protein 12 (nsp12)
subunit of the RNA-dependent RNA polymerase (RdRp). No evidence of global widespread of
RDV-resistance mutations has been reported. Determining emergent mutations prior or
subsequent antiviral therapy has strong implications for clinical management and virus
surveillance.

This study identified 57/149 (38.3%) patients who did not respond to one course (5-days)
(63.2%) or prolonged (5-20 days) (36.2%) RDV therapy by subgenomic RNA detection. Genetic
variants in the *nsp12* gene were detected in 17/49 (34.7%) non-responder patients by next-
generation sequencing, including the *de novo* E83D mutation that emerged in an
immunosuppressed patient after receiving 10+8 days of RDV, and the L838I detected at
baseline and/or after prolonged RDV treatment in 8/49 (16.3%) non-responder subjects.
Although 3D protein modelling predicted no-interference with RDV, the amino acid
substitutions detected in the nsp12 involved changes on the electrostatic outer surface and in
secondary structures that may alter antiviral response.

It is important for health surveillance to study potential mutations associated to drug
resistance and the benefit of RDV retreatment, especially in immunosuppressed patients and
in those with persistent replication.

**Importance**

This study provides clinical and microbiologic data of an extended population of hospitalized
patients for COVID-19 pneumonia who experienced treatment failure, detected by the
presence of subgenomic RNA. The genetic variants found in the *nsp12* pharmacological target
of remdesivir bring into focus the importance of monitoring emergent mutations, one of the
objectives of the World Health Organization (WHO) for health surveillance. These mutations

become even more crucial as remdesivir keeps being prescribed and new molecules are being
repurposed for the treatment of COVID-19.

The present article offers new perspectives for the clinical management of non-responder
patients treated and retreated with RDV, and emphasizes the need of further research of the
benefit of combinatorial therapies and RDV retreatment, especially in immunosuppressed
patients with persistent replication after therapy.

**Keywords:** COVID-19, remdesivir, resistance mutations, subgenomic RNA, retreatment

**Word count:** 200 abstract; 2492 full-text

**BACKGROUND**

The global pandemic of novel coronavirus disease 2019 (COVID-19) caused by severe acute
respiratory syndrome coronavirus 2 (SARS-CoV-2) has created an urgent effort to repurpose
antiviral inhibitors to control viral replication and improve clinical outcomes [1]. Remdesivir
(RDV) was originally developed in response to the 2014-2016 Ebola outbreak in West Africa [2]
and has shown broad-spectrum activity *in vitro* and *in vivo* against pathogenic human
coronaviruses, including the novel SARS-CoV-2 [3]. RDV was the first drug to be approved by
the FDA in October 2020 and has been extensively used in clinical practice during the COVID-
19 pandemic in hospitalized patients [4]. Recently, two oral prodrugs have also been approved
for treatment of COVID-19 patients: molnupiravir and nirmatrelvir/ritonavir [5, 6].

RDV, formerly GS-5734, is a nucleoside analogue pro-drug that inhibits the non-structural
protein 12 (nsp12) subunit of the RNA-dependent RNA polymerase (RdRp) by competing with
its usual natural substrate adenosine triphosphate [7]. The nucleoside analog is incorporated
into the generating RNA strand and evades proofreading to successfully inhibit viral RNA
synthesis. Several clinical trials and a recent control-case study demonstrated reduced time to
recovery, hospitalization time, morbidity and mortality [8-10].

One of the major concerns for health surveillance is determining emergent mutations that
could be associated to drug resistance, fitness advantage, immune escape, or better adaption to
the host. Only few studies have attempted to characterized amino acid substitutions that could
confer resistance to RDV by *in silico* prediction, *in vitro* or in animal models [7, 11-15]. For
instance, F480L, V557L and E802D has been described to conferred 2.4x, 5.2x, 6x-fold decrease
sensitivity to RDV *in vitro*, respectively [7, 14]. Moreover, E802D was recently detected in an
immunocompromised patient after RDV therapy [15], however, no other resistant clinical case
or evidence of global widespread transmission of RDV-resistant mutants after treating with
RDV for over a year have been described thus far.

This study aimed to identify novel genetic variations in the *nsp12* gene in clinical samples
before and after RDV therapy in severe COVID-19 patients who did not respond to therapy
with RDV.

**MATERIALS**

**Study population**

This was an observational prospective study that included 149 COVID-19 patients admitted in
the Hospital Clinic of Barcelona (Spain), from February 2021 until November 2021, who filled
the criteria to receive RDV according to the recommendations of the Spanish Medicine Agency.
These criteria were the following: (1) SARS-CoV-2 positivity confirmed by RT-PCR, (2) ≤ 7 days
from symptoms onset, (3) radiological signs of pneumonia, (4) requiring supplemental oxygen
support or respiratory rate ≥ 24 breaths per minute or $\text{PaO}_2 / \text{FiO}_2 < 300$ mmHg. The RDV dose
used was 200 mg as a loading dose the first day and 100 mg/24h for the next 4 consecutive
107 days. Some immunosuppressed patients received prolonged remdesivir therapy ($> 5\text{d-RDV}$),
decision made by the physician in charge according to the clinical evolution and the immune
status of the patient.

Nasopharyngeal/throat swabs were collected before 1st RDV dose and after the last dose for
each patient. Both samples were tested for SARS-CoV-2 genomic (gRNA) and subgenomic RNA
(sgRNA) by reverse transcriptase-real time polymerase chain reaction (RT-PCR). Although there
is some controversy concerning the use of sgRNA to detect active viral replication, we
previously validated this technique with viral culture [16]. Non-responder subjects were
considered when sgRNA was detected at the end of the RDV treatment. Nucleotide changes
were determined by next generation sequencing (NGS) in pre-and post-RDV treatment clinical
samples of non-responders. Novel amino acid substitutions found in clinical isolates were
evaluated in 3D modelling structure of the protein *in silico*. This study design is showed in
figure 1.

Clinical data were collected to study the overall population, considering the following
variables: age, significant comorbidities, days of treatment with RDV, Intensive Care Unit (ICU)
admission, and all-cause mortality.

**RT-PCR for genomic and subgenomic RNA detection for SARS-CoV-2**

The presence of SARS-CoV-2 gRNA was determined by real time RT-PCR in the automatic
system Cobas 6800 (Roche, Barcelona) according to the manufacturer's instructions.

Inactivation was performed using 1:1 volume of Cobas Omni Lys (Roche, Germany) and total
nucleic acid extraction was done using MagNA Pure Compact (Roche, Switzerland). Elutes
were used for sgRNA test and NGS. *Envelope (E)* sgRNA was detected by real-time RT-PCR
following the procedure previously described [16]. Throat/nasopharyngeal swabs and elutes
were aliquoted and stored at -80°C since their testing. Cycle threshold (Ct) values >40 for gRNA
or sgRNA RT-PCRs were considered negative.

**Next Generation Sequencing of SARS-CoV-2 complete genome**

Retrospectively, the eluted RNA of the SARS-CoV-2 were retrotranscribed into cDNA and SARS-
CoV-2 complete genome amplification was conducted following the openly available protocol
developed by the ARTIC network [17] using Illumina platform. The sequences obtained went
through a bioinformatic pipeline based on a previously described open-source pipeline [18].

Mutations were identified by aligning the consensus sequence with the Wuhan-Hu-1 reference
genome [GenBank: MN908947.3] using Nextclade v.1.14.0. Quality of the sequences were
determined by the quality control metrics of this tool.

**Lineage identity**

The lineage of SARS-CoV-2 was identified using Nextclade v.1.14.0 according to the amino acid
replacements determinant of each variant as classified by the SARS-CoV-2 Interagency Group
(SIG) and the Centres of Diseases Control and Infection (CDC) [19].

**Molecular modelling of mutations in the SARS-CoV-2 nsp12 and spike proteins**

Mutations found genotypically in the nsp12 subunit of the RdRp were generated in the ribbon
structure of the cryo-EM model of the replicating SARS-CoV-2 polymerase complex (PDB 6YY7)
using PyMOL Molecular Graphics System (Schrödinger, version 2.5.2). This software was also
used for structural visualization. Adaptive Poisson-Boltzmann Solver (APBS) Tool 2.1
Electrostatic Plugin included in PyMOL was used for macromolecular electrostatics calculations
and to display the results as an electrostatic potential molecular surface.

**Ethical approval**

The Ethical Committee of our institution accepted the protocol (HCB/2021/0080) and the
included patients signed the informed consent to participate in the study.

**RESULTS**

In a cohort of 149 patients hospitalized for COVID-19 pneumonia, 111 received a 5-dose course
of RDV (5d-RDV) and 38 received prolonged RDV therapy (>5d-RDV) (figure 1). The median age
of the 5d-RDV subset was 62.7 (54; 73) years-old, 81 (73%) presented at least one comorbidity,
34 (30.6%) were admitted in the ICU and 7 (6.3%) died by all-cause mortality (table 1). The
median age in the >5d-RDV subgroup was 59 (53; 67) years-old and they were treated with
RDV for a median of 10 (7; 21) days. Of the 38 patients, 33 (86.8%) had at least one
comorbidity, 19 (50%) were admitted in the ICU and 8 (21.1%) died by all-cause mortality, but
3 deaths were not related to COVID-19. Extended clinical data of both subsets is shown in table
1.

SARS-CoV-2 sgRNA identified 57/149 (38.3%) patients with actively replicating virus after
treatment, classified as non-responders (figure 1). There were 36/57 (63.2%) non-responders
in the 5d-RDV subgroup and 21/57 (36.8%) from the >5d-RDV subgroup. The 17 remaining
patients of the >5d-RDV cohort were not tested for sgRNA and were discarded because the

pre- or post-treatment swab could not be recovered. Ct values corresponding to the gRNA RT-
PCR were maintained or decreased during treatment in non-responder patients. Nucleotide
changes in the *nsp12* gene were detected by NGS and were compared before and after RDV
therapy in non-responder patients. Sequencing quality was achieved in both samples in 28/34
patients of the 5d-RDV subset, and in all 21 patients of the >5d-RDV subset.

NGS allowed the detection of 18 *nsp12* nucleotide substitutions detected in 17/28 (60.7%)
patients from the 5d-RDV subgroup and in 12/21 (57.1%) patients receiving prolonged RDV
therapy. Of the 18 nucleotide substitutions identified, 9 conferred an amino acid substitution,
pointing genetic evolution after treatment (table 2). Mutations were either detected at
baseline (pre-RDV), before and after treatment (pre/post-RDV) or emerged *de novo* after
treatment (post-RDV). The only non-synonym mutation only detected after treatment was
E83D. E83D emerged in a patient (patient 7, table S1) with diffuse large B-cell lymphoma that
was treated with R-CHOP combination chemotherapy and with several antiviral drugs
(lopinavir, ritonavir, hydroxychloroquine, azithromycin, RDV) and convalescent plasma. E83D
mutation emerged *de novo* after 2 courses (10 + 8 days) of RDV. At the end, this patient was
admitted in the ICU and died after 8 months of SARS-CoV-2 infection. The most frequent
mutation (18.4%) was L838I, found at baseline in 1 patient and after prolonged treatment in 8
non-responder subjects. Clinical and genetic data of all patients with *de novo* mutations is
shown in table S1.

NGS allowed the determination of SARS-CoV-2 lineages. Alpha (B.1.1.7) variant was mainly
detected in subjects with 5d-RDV (n=19) compared to >5d-RDV ones (n=3), whereas delta
(B.1.167) was more predominant in >5d-RDV subjects (n=12) than in the 5d-RDV subgroup
(n=4). The variant B.1.177 emerged as an outbreak in Spain during the summer of 2020, and its
incidence was similar in both groups (5d-RDV n=5; >5d-RDV n=6).

**Predicting phenotype of novel *nsp12* mutations by 3D protein modelling**

The position of all genetic variants detected at any time of the treatment were located in the
gene structure of the nsp12 (figure 2) and non-synonym mutations in a 3D protein structure of
the RdRp (figure 3A). *In silico* studies postulated that RDV binds strongly to the active pocket of
the nsp12 by electrostatic interaction with residues R553, R555, T556, K551, W617, D618,
Y619, D623, S682, N691, D760, D761, A762, W800, E811, F812, K813, and S814, and by Van
der Waals bonds in the K621, K622, D623, and L758 [11, 12]. Our genotypic results showed the
emergence of mutations located next to residues involved in RDV-nsp12 binding (E729D,
D738Y, L838I), in the nsp8-nsp12 interaction region (P227L), and inside the RNA-binding
domain of the polymerase (T422I). However, RdRp 3D protein modelling shows that all novel
amino acid substitutions are distant from the RNA binding domain, where the RDV inhibits the
polymerase activity. Replacing one amino acid of a helix for another can change repartition of
amino acids exposed to solvent, as found for the E729D (figure 3B). Slight changes in
conformation were detected for the remaining amino acid substitution.

Most mutations changed a negative charged amino acid to aliphatic-chained ones, substituting
the highly negative electrostatic outer surface of the nsp12 to a major overall neutral status of
the surface (figure 3C). Changes in polarity, charge, and size of the amino acids could
potentially modify interactions between nsp12 and antivirals.

**DISCUSSION**

The study of sgRNA allowed the identification of 57/149 (38.3%) patients who did not respond
to 5-days or >5-days of RDV therapy. Eighteen genetic variants in the *nsp12* gene were
frequently detected in 17/49 (34.7%) by NGS in non-responder subset. No significant viral
mutations were determined to be associated to the failure to RDV treatment, except for the *de*
*novo* E83D mutation that emerged after receiving 18 days of RDV, and the L838I mutation,
which was found at baseline in 1 patient and after prolonged treatment in 8 (18.4%) non-
responders, and its localized next the previously E802D RDV-resistant mutation.

Genetic variants in *nsp12* arose similarly after 5 days of RDV or longer treatment
independently of the duration of therapy (5d-RDV 57.1% vs. >5d-RDV 60.7%) and, surprisingly,
5d-RDV subjects presented more *de novo* mutations (n=6) than >5d-RDV (n=3). Almost all
mutations were detected at baseline in at least one subject showing an evolutionary tuning of
the viral proteins to a new host, although a response for antiviral selective pressure cannot be
excluded in those that persist after treatment. In non-responder patients, viral loads
progressively increased during treatment, as Ct values were maintained/decreased and sgRNA
remained detectable, which could be due to either slow viral shedding or failure to treatment.

Despite the high frequency of substitutions in *nsp12*, none of them have been previously
described to confer resistance to RDV [15]. We detected the L838I mutant, nearby the E802D
RDV-resistant mutation, in 8 patients before and after treatment, and in 1 patient only post-
therapy. All of them received either 10, 20 or 21 days of RDV therapy, 6 of them were
admitted in the ICU and 3 died, thus, it seems to be associated to a worse prognosis, although
the number of cases is scarce to confirm it. It suggests that this naturally occurring variant may
provide an improved viral scape of the inhibitor, since no evidence that RDV acts as a mutagen
driving spontaneous mutations has been reported and the template stalling action of the RDV
limits spontaneous mutations emergence [7, 20].

E83D emerged in the SARS-CoV-2 delta variant that infected an immunosuppressed patient,
who was admitted in the ICU and died after 9 months of infection. The E83D mutation was
detected after RDV retreatment (10 + 8 days), but not before, making it potential for
phenotyping. 3D protein modelling did not predict any interference in the interaction with
RDV, however, its implication in fitness advantage, inhibitor scape or adaption to the host is
unknown.

E729D and D738Y may also have an implication of antiviral response, as they were located in
the palm subunit of the polymerase active site next to the residues involve in the RDV-*nsp12*

interaction and E729D also modified an alpha-helix structure. T422I is located in the conserve
motif G in charge of the RNA template attachment. It is close to the previously described
resistant mutations *in vitro* (F480L, V557L) and *in vivo* in an immunocompromised patient with
persistent viremia (D484Y) [7, 21]. They did not alter the RdRp catalytic site but are thought to
impact RdRp fidelity checking step before catalysis. Molecular surveillance of this region in
RDV-treated COVID-19 patients is suggested to be warranted [20]. Even though 3D protein
modelling predicted none of the mutations found in this study block the binding pocket of
RDV, they involved changes on the electrostatic outer surface and in secondary structures that
may alter antiviral response.

Lineages could be a concern of the severity of the disease and the antiviral response. Alpha
variant was more frequent in the 5d-RDV and delta in the >5d-RDV, which agrees with its
higher pathogenicity. Besides the possible bias caused by the time of inclusion of the patients,
delta lineage could influence on a worse response to RDV and the need of a second course of
treatment.

Available literature about RDV retreatment only reported few clinical cases [22-23] but no
large studies have been carried out, making this practice still unaddressed in current treatment
guidelines [24]. However, this study provides further information of the response to RDV
treatment and retreatment in patients whose active replication was previously checked by
sgRNA and clinical data was exposed.

In conclusion, no significant virological resistance was determined after different courses of
RDV in non-responders severe COVID-19 patients and the duration of RDV treatment does not
seem to be a risk factor for developing RDV-resistance mutations. However, mutations found
in this study, especially E83D, were potential to be further evaluated by recombinant
phenotyping. It is crucial to monitor antiviral resistance as one of the objectives of the World
Health Organization (WHO) for health surveillance, and to study the potential benefit of

combinatorial therapies and RDV retreatment, especially in immunosuppressed patients or
with persistent replication.

**Funding**

This work was financed by a Gilead Sciences grant (IN-ES-540-6089). This work was financed by
ad hoc patronage funds for research on COVID-19 from donations from citizens and
organizations to the Hospital Clínic de Barcelona-Fundació Clínic per a la Recerca Biomèdica.

**REFERENCES**

- 1. Wu Z, McGoogan JM. Characteristics of and important lessons from the coronavirus
disease 2019 (COVID-19) outbreak in China: summary of a report of 72314 cases from
the Chinese Center for Disease Control and Prevention. JAMA 2020;323:1239.
- 2. Mulangu S, Dodd LE, Davey RT Jr, et al. A randomized, controlled trial of Ebola virus
disease therapeutics. N Engl J Med 2019;24:2293–303.
- 3. McCreary EK, Pogue JM. Coronavirus disease 2019 treatment: a review of early and
emerging options. Open Forum Infect Dis 2020;7:ofaa105.
- 4. U.S. Food and Drug Administration. Coronavirus (COVID-19) update: FDA Approves
First Treatment for COVID-19. (2020 October 22) Available from:
[https://www.fda.gov/news-events/press-announcements/fda-approves-first-](https://www.fda.gov/news-events/press-announcements/fda-approves-first-treatment-covid-19)
[treatment-covid-19.](https://www.fda.gov/news-events/press-announcements/fda-approves-first-treatment-covid-19)
- 5. U.S. Food and Drug Administration. Coronavirus (COVID-19) update: FDA authorizes
additional oral antiviral for treatment of COVID-19 in certain adults. U (2021,
December 23). Available from: [https://www.fda.gov/news-events/press-](https://www.fda.gov/news-events/press-announcements/coronavirus-covid-19-update-fda-authorizes-additional-oral-antiviral-treatment-covid-19-certain)
[announcements/coronavirus-covid-19-update-fda-authorizes-additional-oral-antiviral-](https://www.fda.gov/news-events/press-announcements/coronavirus-covid-19-update-fda-authorizes-additional-oral-antiviral-treatment-covid-19-certain)
[treatment-covid-19-certain.](https://www.fda.gov/news-events/press-announcements/coronavirus-covid-19-update-fda-authorizes-additional-oral-antiviral-treatment-covid-19-certain)

- 6. U.S. Food and Drug Administration. Coronavirus (COVID-19) update: FDA authorizes
first oral antiviral for treatment of COVID-19. (2021a, December 22). Available from:
[https://www.fda.gov/news-events/press-announcements/coronavirus-covid-19-
update-fda-authorizes-first-oral-antiviral-treatment-covid-19](https://www.fda.gov/news-events/press-announcements/coronavirus-covid-19-
295 update-fda-authorizes-first-oral-antiviral-treatment-covid-19)
- 7. Agostini ML, Andres EL, Sims AC, et al. Coronavirus susceptibility to the antiviral
remdesivir (GS-5734) is mediated by the viral polymerase and the proofreading
exoribonuclease. *MBio* 2018;9:e00221–18.
- 8. Beigel JH, Tomashek KM, Dodd LE, Mehta AK, Zingman BS, Kalil AC, et al.. Remdesivir
for the Treatment of Covid-19—Final Report. *N Engl J Med*. 2020. [cited 2 Jan 2021].
doi: 10.1056/NEJMoa2007764
- 9. Wang Y, Zhang D, Du G, Du R, Zhao J, Jin Y, et al.. Remdesivir in adults with severe
COVID-19: a randomised, double-blind, placebo-controlled, multicentre trial. *The
Lancet*. 2020;395: 1569–1578. doi: 10.1016/S0140-6736(20)31022-9
- 10. Boglione L, Dodaro V, Meli G, Rostagno R, Poletti F, Moglia R, Bianchi B, Esposito M,
Borrè S. Remdesivir treatment in hospitalized patients affected by COVID-19
pneumonia: A case-control study. *J Med Virol*. 2022 Apr 11:10.1002/jmv.27768. doi:
10.1002/jmv.27768. Epub ahead of print. PMID: 35411627; PMCID: PMC9088403.
- 11. Eweas AF, Alhossary AA, Abdel-Moneim AS. Molecular Docking Reveals Ivermectin and
Remdesivir as Potential Repurposed Drugs Against SARS-CoV-2. *Front Microbiol*. 2021
Jan 25;11:592908. doi: 10.3389/fmicb.2020.592908. PMID: 33746908; PMCID:
PMC7976659.
- 12. Khan FI, Kang T, Ali H, Lai D. Remdesivir Strongly Binds to RNA-Dependent RNA
Polymerase, Membrane Protein, and Main Protease of SARS-CoV-2: Indication From
Molecular Modeling and Simulations. *Front Pharmacol*. 2021 Jul 7;12:710778. doi:
10.3389/fphar.2021.710778. PMID: 34305617; PMCID: PMC8293383.

- 13. Williamson BN, Feldmann F, Schwarz B, Meade-White K, Porter DP, Schulz J, et al.
Clinical benefit of remdesivir in rhesus macaques infected with SARS-CoV-2. *Nature*.
2020;585: 273–276. doi: 10.1038/s41586-020-2423-5.
- 14. Szemiel AM, Merits A, Orton RJ, MacLean OA, Pinto RM, Wickenhagen A, Lieber G,
Turnbull ML, Wang S, Furnon W, Suarez NM, Mair D, da Silva Filipe A, Willett BJ,
Wilson SJ, Patel AH, Thomson EC, Palmarini M, Kohl A, Stewart ME. In vitro selection of
Remdesivir resistance suggests evolutionary predictability of SARS-CoV-2. *PLoS Pathog*.
2021 Sep 17;17(9):e1009929. doi: 10.1371/journal.ppat.1009929. PMID: 34534263;
PMCID: PMC8496873
- 15. Gandhi S, Klein J, Robertson AJ, Peña-Hernández MA, Lin MJ, Roychoudhury P, Lu P,
Fournier J, Ferguson D, Mohamed Bakhsh SAK, Catherine Muenker M, Srivathsan A,
Wunder EA Jr, Kerantzas N, Wang W, Lindenbach B, Pyle A, Wilen CB, Ogbuagu O,
Greninger AL, Iwasaki A, Schulz WL, Ko AI. De novo emergence of a remdesivir
resistance mutation during treatment of persistent SARS-CoV-2 infection in an
immunocompromised patient: a case report. *Nat Commun*. 2022 Mar 17;13(1):1547.
doi: 10.1038/s41467-022-29104-y. PMID: 35301314; PMCID: PMC8930970.
- 16. Santos Bravo M, Berengua C, Marín P, Esteban M, Rodriguez C, Del Cuerpo M, Miró E,
Cuesta G, Mosquera M, Sánchez-Palomino S, Vila J, Rabella N, Marcos MA. Viral
culture confirmed SARS-CoV-2 subgenomic RNA value as a good surrogate marker of
infectivity. *J Clin Microbiol*. 2021 Oct 20;. doi: 10.1128/JCM.01609-21. PMID:
34669457.
- 17. Quick, J. (2020, August 25). NCoV-2019 Sequencing Protocol V3 (locost). protocols.io.
Retrieved May 11, 2022, from <<https://www.protocols.io/view/ncov-2019-sequencing-protocol-v3-locost-bh42j8ye>>
- 18. FISABIO-NGS / SARS-cov2-mapping · GITLAB. GitLab. Retrieved May 11, 2022, from
<https://gitlab.com/fisabio-ngs/sars-cov2-mapping>

- 19. Centers for Disease Control and Prevention. SARS-CoV-2 variant classifications and
definitions. Retrieved May 11, 2022, from <[https://www.cdc.gov/coronavirus/2019-ncov/variants/variant-](https://www.cdc.gov/coronavirus/2019-ncov/variants/variant-classifications.html#:~:text=On%20November%2030%2C%202021%2C%20the,among%20those%20without%20travel%20history)
[classifications.html#:~:text=On%20November%2030%2C%202021%2C%20the,among](https://www.cdc.gov/coronavirus/2019-ncov/variants/variant-classifications.html#:~:text=On%20November%2030%2C%202021%2C%20the,among%20those%20without%20travel%20history)
[%20those%20without%20travel%20history](https://www.cdc.gov/coronavirus/2019-ncov/variants/variant-classifications.html#:~:text=On%20November%2030%2C%202021%2C%20the,among%20those%20without%20travel%20history)>.
- 20. Lo MK, Albariño CG, Perry JK, Chang S, Tchesnokov EP, Guerrero L, et al. Remdesivir
targets a structurally analogous region of the Ebola virus and SARS-CoV-2 polymerases.
*Proc Natl Acad Sci.* 2020;117: 26946–26954.
- 21. Martinot M, Jary A, Fafi-Kremer S, Leducq V, Delagreverie H, Garnier M, Pacanowski J,
Mékinian A, Pirenne F, Tiberghien P, Calvez V, Humbrecht C, Marcelin AG, Lacombe K.
Emerging RNA-Dependent RNA Polymerase Mutation in a Remdesivir-Treated B-cell
Immunodeficient Patient With Protracted Coronavirus Disease 2019. *Clin Infect Dis.*
2021 Oct 5;73(7):e1762-e1765. doi: 10.1093/cid/ciaa1474. PMID: 32986807; PMCID:
PMC7543308.
- 22. Helleberg M, Niemann CU, Moestrup KS, Kirk O, Lebech AM, Lane C, Lundgren J.
Persistent COVID-19 in an Immunocompromised Patient Temporarily Responsive to
Two Courses of Remdesivir Therapy. *J Infect Dis.* 2020 Sep 1;222(7):1103-1107. doi:
10.1093/infdis/jiaa446. PMID: 32702095; PMCID: PMC7454684.
- 23. Choi B, Choudhary MC, Regan J, et al. Persistence and evolution of SARS-CoV-2 in an
immunocompromised host. *N Engl J Med.* DOI: 10.1056/NEJMc2031364
- 24. Al-Heeti O, Kumar RN, Kling K, Angarone M, Achenbach C, Taiwo B. Remdesivir
retreatment: another unproven intervention for COVID-19. *J Antimicrob Chemother.*
2022 Feb 23;77(3):854-856. doi: 10.1093/jac/dkab472. PMID: 35022726; PMCID:
PMC8865007.
- 25. Gao Y, Yan L, Huang Y, Liu F, Zhao Y, Cao L, Wang T, Sun Q, Ming Z, Zhang L, Ge J, Zheng
368 L, Zhang Y, Wang H, Zhu Y, Zhu C, Hu T, Hua T, Zhang B, Yang X, Li J, Yang H, Liu Z, Xu

369 W, Guddat LW, Wang Q, Lou Z, Rao Z. Structure of the RNA-dependent RNA
polymerase from COVID-19 virus. Science. 2020 May 15;368(6492):779-782. doi:
10.1126/science.abb7498. Epub 2020 Apr 10. PMID: 32277040; PMCID: PMC7164392.

**FIGURES**

**Figure 1. Scheme of the study design.** Patients positive for SARS-CoV-2 that were admitted in
the Hospital Clinic of Barcelona (Spain) for COVID-19 pneumonia and were treated with
remdesivir (RDV) were included in the study. They were treated with 5 doses (1 course) of RDV
or with longer treatments (>5 doses). Subgenomic RNA (sgRNA) detection was performed in all
samples in order to detect viral replication before and after treatment. Patients with positive
sgRNA after treatment were classified as non-responders and were sequenced by next-
generation sequencing (NGS). Not all clinical samples could be sequenced with enough quality
to be analysed and included in the study due to RNA degradation or low viral load in the
sample. Of the samples from the >5-dose subpopulation that could be studied, all were sgRNA
RT-PCR positive in the last sample. N indicates the number of subjects included in each cohort.

**Figure 2. Gene structure of the SARS-CoV-2 non-structural protein 12 with the novel**
**mutations detected.** The nucleotide position is indicated respect to Wuhan-Hu-1 reference
genome [GenBank: MN908947.3]. Amino acid substitutions are indicated in brackets. Gene
structure is based on Gao et al, 2020 [25].

**Figure 3. Location of novel RdRp mutations in 3D protein models.** (A) Theoretical structure of
RdRp is represented in the ribbon structure of the cryo-EM model using PyMOL. The different
subunits of the RdRp are colored as follow: nsp7 in blue, nsp8a in yellow, nsp8b in pink, nsp12
in green. The RNA duplex is colored in orange and blue. Novel genotypically detected
mutations indicated in red, in the right figure. (B) E729D mutation breaks the alpha-helix
secondary structure of the nsp12 (C) APBS-generated electrostatic surface of the RdRp.
Negative charged areas are indicated in red and positive charged areas in blue. Mutations are
visualized in red and tagged in the figure.

A

B

C

**TABLES**

**Table 1. Clinical characteristics of the study population according to the treatment of**
 **remdesivir received.**

Clinical characteristics	5d RDV	>5d RDV
N	111	38
Age (median; IQR)	62.7 (54; 73)	59 (56; 67)
Days of RDV therapy (median; IQR)	5	10 (7; 21)
Comorbidities (n,%)	81 (73%)	33 (86.8%)
• Hypertension (n,%)	51 (45.9%)	8 (21.1%)
• Diabetes mellitus (n,%)	26 (23.4%)	5 (13.2%)
• Obesity (n,%)	18 (16.2%)	3 (7.9%)
• Cardiovascular disease (n,%)	32 (28.8%)	3 (7.9%)
• chronic pulmonary disease ^a (n,%)	24 (21.6%)	6 (15.8%)
• chronic kidney failure (n,%)	9 (8.1%)	1 (2.6%)
• haematological malignancies ^b (n,%)	15 (13.5%)	25 (65.8%)
• Solid malignancy with active chemotherapy (n,%)	4 (3.6%)	1 (2.6%)
• Transplant recipients (n,%)	4 (3.6%)	6 (15.8%)
• Other disorders treated with immunosuppressors (n,%)	7 (6.3%)	1 (2.6%)
ICU admission (n,%)	34 (30.6%)	21 (55.3%)
Mortality (n,%)	7 (6.3%)	8 (21.1%)

412 N: number of subjects

Age and days of treatment are indicated as the median and the interquartile range Q1; Q3.

414 ^a Chronic pulmonary disease includes chronic obstructive pulmonary disease and asthma

415 ^b Haematological malignancies include lymphoma or leukaemia

Abbreviations: RDV remdesivir, ICU Intensive Care Unit

**Table 2. *Nsp12* nucleotide substitutions detected at baseline (pre-RDV), at baseline and after**
 **treatment with remdesivir (pre/post-RDV) and after (post-RDV) therapy in clinical isolates.**

Mutations ^a	Location	Pre-RDV ^b	Pre/post-RDV ^b	Post-RDV ^b	ICU ^b	Mortality ^b
A13535G (Y32C)		2	1 (10d)	1 (5d)	1	0
C13551T				1 (18d)	1	1
G13564T (V42L)			1 (5d)		0	0
A13689T (E83D)				1 (18d)	1	1
A13711G (K91G)			1 (5d)		0	0
C14119T	Nsp8-interaction		1 (5d)		0	0
C14120T (P227L)	Nsp8-interaction		3 (5d n=2, 10d)	1 (5d)	3	1
C14178T	Nsp8-interaction		1 (5d)		0	0
C14547A	Nsp7-8-interaction		2 (5d)	3 (5d)	3	0
C14703T (T422I)	RNA binding site (motif G)		1 (10d)		1	0
C15237T	RNA binding site (motif B)			1 (10d)	0	0
C15240T	RNA binding site (motif B)		1 (5d)		1	1
C15324T	RNA binding site	2	1 (30d)	2 (5d)	3	1
C15441T	RNA binding site (motif C)		1 (5d)		0	0
G15627T (E729D)	RNA binding site (motif E)		1 (5d)		1	0
C15652T (D738Y)	RNA binding site		1 (5d)	1 (5d)	1	0
G15910T			1 (10d)		0	0
C15952A (L838I)			8 (10d n=7; 21d)	1 (20d)	6	3

420 ^a The nucleotide position is indicated respect to Wuhan-Hu-1 reference genome [GenBank: MN908947.3]. Amino
 acid substitution is indicated in brackets.

422 ^b Indicate the number of subjects with the specific substitution, admitted in the ICU or died due to all-cause
mortality. Days of treatment received when the mutation was found are indicated in brackets.
Abbreviations: Nsp non-structural protein, RDV remdesivir, ICU intensive care unit.

**SUPPLEMENTARY DATA**

**Table S1. Clinical characteristics of the patients with SARS-CoV-2 non-structural protein 12 and/or Spike mutations of interest.**

ID	Age	Comorbidity	ICU admission	Mortality	Immunosuppressive treatment	Antiviral treatment	Date of infection	Linage	NSP12	
									Before RDV	After RDV
60	Hypertension, enolic dilated myocardopathy	no	no	BNB	RDV 5d	March 2021	Alpha (B.1.1.7)	-	A13535G (Y32C), C14120T (P227L), C15324T
68	Hypertension	yes	no	TCZ, CORT	RDV 5d	April 2021	Alpha (B.1.1.7)	C14120T (P227L)	C14120T (P227L), C15324T
40	No	no	no	BNB	RDV 5d	March 2021	Alpha (B.1.1.7)	-	G14547A
47	No	no	no	TCZ, BNB, CORT	RDV 5d	April 2021	Alpha (B.1.1.7)	A13535G (Y32C), C14120T (P227L), C15324T	G14547A
40	No	yes	no	TCZ, BNB, CORT	RDV 5d	April 2021	Delta (B.1.167)	A13535G (Y32C), C14120T (P227L), C15324T	G14547A
49	Thalassemia minor	no	no	BNB	RDV 5d	June 2021	Alpha (B.1.1.7)	-	G15652T (D738Y)
83	Diffuse large B-cell lymphoma	yes	yes	R-CHOP ^a	LPV + RTV + HCQ 7d, AZM 5d, RDV 10d + 8d, plasma	March – December 2021	Delta (B.1.167)	-	C13551T, A13689T (E83D)
67	Kidney transplant, arterial hypertension, hypercholesterolemia	no	no	BNB 10d	RDV 10d	July 2021	Delta (B.1.167)	G15910T	C15237T, G15910T
64	Mantle lymphoma in complete remission	yes	No	DEX 10d, TCZ, BNB, anakinra	RDV 20d, TEC, IVM + plasma	August 2021	Delta (B.1.167)	-	C15952A (L838I)

427 ^a R-CHOP is a chemotherapy composed by the combination of rituximab, cyclophosphamide, hydroxidaunorubicine, oncovin and prednisone

Abbreviations: ICU intensive care unit, LPV lopinavir, RTV ritonavir, RDV remdesivir, HCQ hydroxichloroquine, AZM Azithromycin, TEC teicoplanin, IVM ivermectin, TCZ tocilizumab, BNB

baricitinib, DEX dexamethasone, CTX cyclophosphamide, PDN prednisone, CORT other corticoids.

430

Response to reviewers

Reviewer comments:

Reviewer #1 (Comments for the Author):

Dear authors, I would like to thank you for your work. Your work is impressive in this section and gives us some important points about Remdesivir as an antiviral SARS-CoV-2. It might give scientists a new way to cure COVID-19.

These my comments:

Lines (27-28) We need to define genetic and biochemical pathways to RDV resistance and emphasize the need for additional studies to define the potential for emergence of these or other RDV resistance mutations in clinical settings. So you need to edit this paragraph.

We rewrote it with your comments, however, the paragraph indicates the objective of the study that consists on studying genetic variants before and after remdesivir therapy in clinical samples.

In lines (33 and 36) You need to review numbers

Thank you very much for your correction. We change the denominators and the percentages of each subgroup of non-responder patients.

In line (118) Why did not add a gender variable?

Thank you for your comment. We added the gender of both subgroups in table 1 and in the manuscript in lines 156, 157 and 161.

In lines (153-160) There is concern about the death rate, which is high compared to the number of patients who received RDV. Are there drug interactions or the effect of this drug on the chronic diseases of patients and old age? This should be clarified.

The mortality rate of the 5d-RDV subgroup is 6.3% which is not high considering all of them were hospitalized patients for COVID-19 pneumonia and 30% were admitted in ICU. Similarly, the approximately 30% of the >5-d RDV subgroup is not high considering 55% were admitted in ICU. The clinical history of the last subgroup indicates that >85% had comorbidities that are associated with worse outcomes and the majority received immunosuppressive therapy.

There is a warning about possible drug interactions between immunosuppressive drugs and those already approved or under investigation for the treatment of COVID-19. Currently, there is no data on possible interactions between RDV and immunosuppressive drugs, unlike chloroquine/hydroxychloroquine and lopinavir/ritonavir (Li *et al*, 2020).

Li Y, Yang N, Li X, Wang J, Yan T. Strategies for prevention and control of the 2019 novel coronavirus disease in the department of kidney transplantation. *Transplant International*. 2020;33(9).

In lines (215-216) Can you explain this paragraph?

Genetic variants were studied before and after treatment in patients receiving RDV for 5 days and in patients receiving RDV from 6 to 20 days. It was expected that genetic variants emerged more frequently in patients with longer treatments as they had more chance to emerge and be selected by the drug, however, our results showed that mutations were detected equally in both

groups, suggesting it does not depend on the duration of the therapy. This information has been clarified in lines 220-221.

In lines (232-235) You need to add more information to clear up this confusion.

We described the clinical case of the patient where the variant E83D was detected. As this mutation was only detected after RDV therapy, we phenotyped it by 3D protein modelling and its location suggested there is no interference with the mechanisms of action of RDV, but there are multiple implications that the mutation might cause that were not confirmed, such as an altered replicative capacity or a better adaptation to the host by evading the immune system.

In line (401) Figure 2, you need to change the color of words to make them clear to the audience.

Thank you for your comment. I think you meant figure 3 as it is the one corresponding to line 401. We tried to interfere the minimum possible on the figure so the 3D protein structure can be correctly visualized and the localization of the amino acid. However, we will change the background of each tag to black with white font.

No statistical software was used to analyze the results.

There is not statistical analysis in the study that need to be performed. The maximum degree of mathematics data presented are the N (number of subjects or mutations) and the percentage in relation to the total of samples of the group. We used clinical data to describe the clinical history in which the genetic variants were detected, but not to determine factors associated with COVID-19 diseases, that is the reason why we did not show significance either associations.

Reviewer #2 (Comments for the Author):

Bravo and co-workers reported a clinical series of some 100+ covid-19 cases with remdesivir treatment and associated viral whole-genome NGS data. They reported that multiple mutations on the nsp12 (RdRp) gene were found but they were not on or in the vicinity of the known RdRp active site and concluded that no virological resistance was found during short (5-day) and longer (5+) courses of remdesivir therapy. Integrated clinical and virological data are important to the field. But this reviewer identified technical flaws in the study design that should be properly addressed before supporting the conclusion of the present work.

Major

1. This study was not a case-control study and there was no functional validation of the mutations in vitro or in cell line to disprove their association with remdesivir resistance. In addition, the authors have no attempts to provide evidence that those sequences collected with remdesivir therapy has mutation rate comparable to background level (i.e., without remdesivir treatment) to support their claims that those mutations are not remdesivir-driven. As the authors performed WGS, such data should be readily available for comparison with those already reported in the literature. This is critically needed before jumping to any conclusions currently based on homologous modelling only without any functional validation.

This is an uncontrolled before-and-after study or intervention study in which we evaluated the genetic variants before and after treatment in each individual subject, therefore, every mutation is compared with the background of the sample collected previously to remdesivir therapy. We did not compare it with a control subpopulation that has never received remdesivir because

those sequence are publicly available in many repositories. Our results suggested that mutations that only emerged after treatment but were not at baseline in the same subject, and were not reported to be specific of a new lineage, could be driven by remdesivir therapy. Effectively, the results of the WGS are available and we checked that those variants were not reported previously elsewhere.

2. In retrospect, the authors should have collected samples on an ideally daily basis so that those from non-responders could also be sequenced and analysed as long as they remained viral RNA positive and could have served as a comparison group for mutation rate and de novo mutation appearance. The inclusion of another control group comprising COVID-19 patients without remdesivir treatment is also essential. For example, a similar random mutation patterns between treatment and no-treatment groups can indirectly indicate that the mutations were not associated with antiviral resistance. The key message is that proving and disproving association of mutations to antiviral resistance require the same par of evidence that the current study unfortunately lack.

In the design of the study, we firstly thought to include a control group of subsets infected with SARS-CoV-2 that were not treated with remdesivir. However, in our hospital, it was impracticable to have patients without remdesivir treatment, since those who met the criteria received remdesivir since the beginning of the pandemic. Moreover, variants were evolving very fast and each virus had different baseline mutations that needed to be considered in order to appropriately study the selection by the antiviral drug. Therefore, we concluded that the best control we could add is to have a sample from the same patient collected before receiving remdesivir, so the sequencing result could have the same clinical background in every case, with the only study variable of remdesivir therapy.

Minor

3. Line 133: "retrotranscribed" should have read as "reverse-transcribed".

Corrected.

4. Lines 133-136: More technical descriptions on the NGS workflow would be very helpful. Simply saying "as previously described" is far from reader-friendly. At least the authors should provide name of the pipeline and key algorithm used in the analysis NGS data.

Thank you for your comment. The name of the pipeline has been added in the method section (line 136). This is a public pipeline which is available at the following link:

<https://gitlab.com/fisabio-ngs/sars-cov2-mapping/>

The workflow for Illumina data is based on iVar and consists of the following steps:

1. Quality trimming
2. Mapping of reads to MN908947.3 reference genome
3. Primer trimming
4. Consensus genome generation
5. Variant calling
6. Basic QC summaries

I think your comment is very helpful, however, we consider this information very technical to be added in the methods section and it could be easily followed in the website referenced in the manuscript.

5. Bioinformatics pipeline and cut-off/ threshold used in the identification of purported mutations associated (or not associated) with remdesivir therapy needs to be clearly described. For example, sequences existed as quasispecies and did the mutations need to be present in 100% of the illumine reads in order to qualify as a mutation? If no, what was the selection criteria and the rationale behind?

The variant was considered in the consensus sequence generated after the bioinformatic analysis when it was present in at least 80% of reads with a minimum quality of 20 and minimum depth per position of 30:

```
MINQUAL_CONS=20 # Minimum quality for consensus calling
```

```
MINFREQ_CONS=0.8 # Minimum frequency to consider fixed a SNP in consensus
```

```
MINDEPTH_CONS=30 # Minimum position depth, ambiguous_char otherwise
```

```
AMBIGUOUS_CHAR=N # Character to use in consensus for uncovered positions
```

This information has been added in lines 137-138 of the manuscript.

6. Figure 1: the "Positive sgRNA" and "Negative sgRNA" labels should indicate that they referred to samples collected at the last day of remdesivir dosing.

Corrected.

7. Lines 157-160: What do the numbers in parenthesis refer to, IQR or range or something else? Please clarify.

It refers to the interquartile range (quartile 1; quartile 3). We corrected in the manuscript.

8. The work would benefit from English-editing.

Thank you for your suggestion, this article was corrected by a native English editor called Donna Pringle. We actually added it in the acknowledgment section (lines 279-280) but we will check again and correct English grammar.

Reviewer #3 (Comments for the Author):

This is a very interesting paper and I feel that it adds to the body of work surrounding COVID-19. Here are some suggestions:

Line 27: Delete 'of', it should read "No evidence of global widespread 28 RDV-resistance mutations has been reported"

Corrected.

Line 28 - 30: Change wording from 'or' into 'to'. "Determining emergent mutations prior to..."

The actual correction should be prior and subsequent antiviral therapy, as we sequence both clinical samples before and after remdesivir therapy. We corrected in the manuscript.

Line 32: Why is there a (63.2%) in the sentence. Needs clarification.

Thank you for your correction. The percentages indicate non-responder patients among the total patients of each subgroup: 36 of 111 patients receiving 5 days of RDV, and 21 of 38 patients receiving between 6-20 days of RDV. We corrected it in the line 33.

Line 33 -34: Clarify Next Generation Sequencing. 454 Pyrosequencing or Illumina, or some other type?

We performed next generation sequencing using the Illumina platform as indicated in the method section, but we now also corrected it in the abstract.

Line 40 - 42: Try not to use 'and' two times in one sentence. Substitute one of the 'and' for 'as well as'

Corrected.

Line 47: Capitalize Remdesivir or use RDV since it was defined earlier.

Thank you for your correction. The name of drugs must be written in lower case letters, however, we substituted it for RDV in this case. Thank you.

Line 49: Capitalize Remdesivir or use RDV since it was defined earlier.

Corrected.

Line 79: Capitalize the drug names. They are proper nouns.

As we mention the name of drugs must be written in lower case letters, they are not proper names and are written in such a way by the FDA.

U.S. Food and Drug Administration. Coronavirus (COVID-19) update: FDA Approves First Treatment for COVID-19. (2020 October 22) Available from: <https://www.fda.gov/news-events/press-announcements/fda-approves-first-treatment-covid-19>.

Line 89: Add a comma after in vitro

Corrected.

Line 90: Add a comma after V557L

Corrected.

Line 116: Clarify Next Generation Sequencing. 454 Pyrosequencing, Illumina, or some other type?

We added in that section that the platform used was Illumina.

Line 157: Clarify what (54;73) means. Is this 54 - 73 years old?

It refers to the interquartile range (quartile 1; quartile 3). We corrected in the manuscript.

Line 178: Remove the second 'only' in the sentence. It should read, "The only non-synonym mutation detected..."

Corrected.

Line 181: Try to avoid using 'and' two times in one sentence. Change the 'and' to 'as well as'

Corrected.

Line 194: Be constant with your italics of nsp12.

Thanks for the correction, we used italics when it refers to the gene sequence, but not when refers to the protein.

Line 196: Be constant with your italics of nsp12.

In this case, it should not be in italics as it refers to the protein and the different residues implicated in the interaction between the nsp12 protein and the drug.

Line 195-198: Run-on sentence. Break into two sentences if possible.

It is a long sentence because we name all residues implicated in the interaction, however, we cannot break it into two sentences as it will lose the sense of the sentence.

Line 207: Be constant with your italics of nsp12.

In this case, it should not be in italics as it refers to the protein.

Line 209: Be constant with your italics of nsp12.

In this case, it should not be in italics as it refers to the protein.

Line 213 - 217: Avoid using 'and' multiple times in a sentence. Break this sentence into 1-3 sentences because it is too long.

Thanks for your comment. I broke it into 3 different sentences.

Line 410: Make sure that in Table 1, all of your Clinical Characters are Capitalized. Be consistent.

Corrected.

September 2, 2022

Dr. Marta Santos Bravo

Microbiology Department, Hospital Clínic I Provincial de Barcelona, University of Barcelona. Institute for Global Health (ISGlobal), Barcelona, Spain.

Microbiology, Hospital Clinic of Barcelona

Villarroel Street, 170.

Barcelona, Barcelona 08036

Spain

Re: Spectrum02448-22R1 (Genetic study of SARS-CoV-2 nsp12 in non-responder COVID-19 patients to remdesivir)

Dear Dr. Marta Santos Bravo:

Your manuscript has been accepted, and I am forwarding it to the ASM Journals Department for publication. You will be notified when your proofs are ready to be viewed.

Sincerely,

Abimbola Kolawole
